

# Model dependence in multi-model climate ensembles: weighting, sub-selection and out-of-sample testing

Gab Abramowitz[1,2], Nadja Herger[1,3], Ethan Gutmann[4], Dorit Hammerling[4], Reto Knutti[5], Martin Leduc[6,7], Ruth Lorenz[5], Robert Pincus[8,9], Gavin A. Schmidt[10]

[1] Climate Change Research Centre, UNSW Sydney, Australia
    [2] ARC Centre of Excellence for Climate Extremes, Australia
    [3] ARC Centre of Excellence for Climate System Science, Australia
    [4] National Center for Atmospheric Research, Boulder, Colorado, USA
    [5] Institute for Atmospheric and Climate Science, ETH Zurich, Switzerland
[6] Ouranos, Montréal, Québec, Canada
    [7] Université du Québec à Montréal, Montréal, Québec, Canada
    [8] Cooperative Institute for Research in Environmental Sciences, University of Colorado, USA
    [9] NOAA Earth System Research Lab, Physical Sciences Division, USA
    [10] NASA Goddard Institute for Space Studies, New York, NY, USA

*Correspondence to*: Gab Abramowitz (gabriel@unsw.edu.au)

**Abstract.** The rationale for using multi-model ensembles in climate change projections and impacts research is often based on the expectation that different models constitute independent estimates, so that a range of models allows a better characterisation of the uncertainties in the representation of the climate system than a single model. However, it is known that research groups share literature, ideas for representations of processes, parameterisations, evaluation data sets and even

sections of model code. Thus, nominally different models might have similar biases because of similarities in the way they represent a subset of processes, or even be near duplicates of others, weakening the assumption that they constitute independent estimates. If there are near-replicates of some models, then treating all models equally is likely to bias the inferences made using these ensembles. The challenge is to establish the degree to which this might be true for any given application. While this issue is recognized by many in the community, quantifying and accounting for model dependence in anything other than

an ad-hoc way is challenging. Here we present a synthesis of the range of disparate attempts to define, quantify and address model dependence in multi-model climate ensembles in a common conceptual framework, and provide guidance on how users can test the efficacy of approaches that move beyond the equally weighted ensemble. In the upcoming Coupled Model Intercomparison Project phase 6 (CMIP6), several new models that are closely related to existing models are anticipated, as well as large ensembles from some models. We argue that quantitatively accounting for dependence in addition to model

performance, and thoroughly testing the effectiveness of the approach used will be key to a sound interpretation of the CMIP ensembles in future scientific studies.

## 1 Characterising uncertainty in ensemble projections

Future climate projections are uncertain for a wide range of reasons. There is limited knowledge of the future of human behaviour (including the greenhouse gas and other emissions associated with it); there are limitations in our understanding of

how the climate system works; in our ability to codify what is understood into models; in our ability to resolve known processes in models due to computational limitations; in the measurements of the state of the climate in the past required for accurate model initialisation; and in the inherent limits of predictability associated with the climate system itself given its chaotic nature. While the first of these issues is addressed by considering projections conditional on specific emission scenarios, the other areas of uncertainty are addressed by the use of multiple working hypotheses (Chamberlin, 1890), in the form of multi-model

ensembles of climate projections.



By using an ensemble of climate projections from a range of climate models, as opposed to a single model or single simulation, researchers hope to achieve two related goals. First, they want simulation agreement across models in the ensemble to imply robustness, for any particular phenomenon of interest or, more generally gain an understanding of the structural uncertainty in any prediction. More ambitiously, they would also like the distribution of behaviour of any particular phenomenon across the

ensemble to be a reasonable approximation of the probability of its occurrence, conditional upon the assumptions described above.

While we described six sources of uncertainty affecting this likelihood in the paragraph above, understanding the relationship between models and the real world is aided by sorting them into just two categories. The first is *epistemic uncertainty*, which

relates to our knowledge and understanding of the climate system, and so encompasses uncertainties that are thought to be reducible with more information or knowledge. If for a moment we assume that the climate system is fundamentally deterministically predictable, and that observational records were spatially complete and long enough to characterise any internal variability, an ideal model ensemble distribution would then accurately represent our uncertainty in creating and using climate models. That is, it would represent uncertainty in our understanding of how the climate system works, our ability to

codify what is understood in models, and our ability to resolve known processes in models due to computational limitations - as noted above. The existence of epistemic uncertainty in climate modelling is inevitable, since they are low dimensional modelling systems that can never be a perfect representation of the climate system (Box, 1979; Oreskes et al, 1994).

Alternatively, if we assume that we did have a perfect understanding of how the climate system worked and could codify this

effectively in models, climate system stochasticity and chaotic behaviour will also mean that for any given (incomplete) observational record and model resolution, an ensemble of solutions would exist, representing the inherent limit of predictability in the climate system - *aleatory uncertainty*.

The use of multi-model ensembles, including those from the widely-used Coupled Model Intercomparison Project (CMIP), is

common in climate science despite the fact that such ensembles are not explicitly calibrated to represent an independent set of estimates of either epistemic or aleatory uncertainty. In fact the ensembles are not systematically designed in any way, but instead represent all contributions from institutions with the resources and interest to participate - and so are optimistically called 'ensembles of opportunity' (Tebaldi and Knutti, 2007). The purpose of this paper is to give an overview of approaches that have been proposed to untangle this ad hoc ensemble sampling strategy, to discuss assumptions behind the different

methods as well as the advantages and disadvantages of each approach. In the end, the goal is to efficiently extract the information relevant to a given projection or impacts question, beyond naive use of CMIP ensembles in their entirety. Identifying appropriate approaches for a given application might not only help to give more understanding and certainty to projections and impacts research, but might also help avoid wasting resources on developing models that essentially duplicate information that other models provide.

**2 Ensemble sampling to address uncertainties**

Despite the fact that state-of-the-art ensembles such as CMIP are not calibrated to provide independent estimates, they do sample both epistemic and aleatory uncertainty, by integrating an arbitrary number of emissions scenarios, global climate models, physics perturbations and initial-condition realizations. This leaves practitioners to interpret the ensemble distribution very much on a case-by-case basis, and decomposing these uncertainties is not necessarily straightforward (see Hawkins and

Sutton 2009, 2011; Leduc et al, 2016a). For instance, if only a subset of the CMIP5 ensemble is considered, the way simulations are selected along the three axes (scenario, model, realisation) will modulate the relative role of each source of uncertainty. This effect is implicit even when considering all available simulations because the broader ensembles such as CMIP are



generally constructed based on an the ability of the international modelling centres to contribute, rather than a systematic sampling strategy (Tebaldi and Knutti, 2007).

In all of these cases, the motivation for using an ensemble rather than a single simulation is to obtain multiple independent estimates of the quantity under consideration. While the weather forecasting community has developed metrics that give useful information about ensemble spread interpreted as aleatory uncertainty (e.g. rank frequency histograms or rank probability skill scores), they remain necessary but not sufficient conditions for assessing the independence of aleatory uncertainty estimates (see Hamill 2001; Weigel et al, 2007). In particular, the effect of epistemic uncertainty on these metrics is difficult to interpret. Generally speaking, there is no established and universally accepted methodology to define whether or not ensemble members
are independent. The implications of a lack of independence are clear: agreement within an ensemble might not imply robustness, and similarly, ensemble distribution is unlikely to provide useful information about the probability of outcomes. Indeed recent work by Herger et al (2018a) showed that sub-selection of ensemble members by performance criteria alone, without consideration of dependence can result in poorer ensemble mean performance than random ensemble member selection. Despite this heuristic understanding of the perils of model dependence, a practical definition of model independence
is not straightforward, and is problem dependent. In reviewing the literature here, we do not aim to provide a single canonical definition of model dependence, but instead contextualise the range of definitions and applications that have been used to date in a single conceptual framework, and reinforce the need for thorough out-of-sample testing to establish the efficacy of any approach for a given application.

**3 What is meant by model independence?**

Though the statistical definition of independence of two events *A* and *B* is strictly defined as *P(A|B) = P(A)* (as dealt with in some depth by Annan and Hargreaves, 2017), it is not immediately clear that there is an objective or unique approach to applying this definition to climate projection ensemble members. Indeed there is an obvious way in which models should be *dependent* - they should all provide a good approximation to the real climate system. Noting that each model is a myriad of discrete process representations makes it clear how complicated any categorical statement about model independence within
an ensemble needs to be. For those process representations where models exhibit high fidelity (i.e. where there is sufficient observational constraint to ascertain this), models should be expected to agree in their representation. That is, where we have clear observational evidence, we do not expect a model to exhibit epistemic departures from the true physical system. It is only in the cases where there is insufficient observational constraint to diagnose such an epistemic departure, or those where no model can avoid one, that models should provide independent process representations (e.g. parameterisation of known
processes because of scale considerations or empirical approximations for complex or incomplete process representations).

It might therefore seem that the limited case of ascertaining whether two models are independent with respect to a specific process representation that is weakly constrained by observational data would be relatively easy to verify. If the two models take different approaches to the under-constrained process, we might argue, they're independent with respect to it. This is
clearly unsatisfactory though. First, we have no context for how different treatments of this process might legitimately be, or the ability to quantify this difference. Next, looking for evidence of the independence of these process representations via the impact they have on simulated climate is also fraught, since we are reliant on the effect they have within a particular modelling system. Two radically different representations of a process might not elicit different responses in a modelling system if the modelling system is insensitive to the process in question. Alternatively, one representation may result in artificially strong
model performance (and so misleadingly imply fidelity) if it effectively compensates for other biases within the modelling system. This is an example of epistemological holism well documented by Lenhard and Winsberg (2010). This problem is further compounded by the reality of models being very large collections of process representations, where only a subset of



these might be independent. In this context, a categorical statement about overall model independence seems far too simplistic. It also means an assessment of dependence will have different outcomes depending on the phenomena and question being investigated, since different parts of a model may affect the problem under consideration.

### 4 Independence as distinct development paths

Understanding the evolutionary history of models might also seem a path to understand and characterise model independence (e.g. Figure 5 in Edwards, 2011). This *a priori* view of independence is analogous to evolutionary cladistics - those models that share common ancestry might be deemed partially dependent. The analogy is of course not perfect, since modelling groups have borrowed discrete components from each other over time, but nevertheless the lineage of any particular model snapshot at a point in time might theoretically be traced. To date, with the exception of Boé (2018) who utilises version number as a

proxy for differences between model components, we are not aware of any studies that comprehensively try to do this, most likely because of the paucity of information on each model's history, including the lack of freely available source code. Boé (2018) accounted for the number of shared components by GCMs and quantified how the replication of whole model components, either atmosphere, ocean, land or sea ice, influences the closeness of the results of different GCMs and could show a clear relationship, where even a single identical component had a measurable effect. This was shown to be true for

global as well as regional results, however, no component appeared to be more important than the others.

While defining model independence *a priori* is desirable, it quickly becomes difficult and time consuming for large ensembles such as CMIP, particularly given the lack of transparency regarding precisely what constitutes a given model, the difficulty deciding whether or not components are identical, and the role of tuning (see Schmidt et al, 2017). Boé (2018) used version

numbers, considering components to be different when the first version number was different, but not if the second version number was different. Since version numbering is inconsistent across modelling centres, two components might be very different even if they share the first version number (for example CLM4 and CLM4.5) or vice versa. Also, it does not seem obvious how we might account for the effect of shared model histories within an ensemble if we had all this information available, beyond categorical inclusion or exclusion of simulations. A reminder that shared history as it pertains to dependence

should only include process representations that are not tightly observationally constrained (so that Navier-Stokes equations might *not* represent dependent process treatment, for example), as discussed above.

### 5 Independence as inter-model distance

Alternatively, dependence could be defined *a posteriori* in terms of the statistical properties of model output (perhaps more analogous with Linnean taxonomy). This is the approach taken by Masson and Knutti (2011) and Knutti et al. (2013), who

used hierarchical clustering of the spatiotemporal variability of surface temperature and precipitation in climate model control simulations to develop a 'climate model genealogy'. Perhaps unsurprisingly, they found a strong correlation between the nature of model output and shared model components. While the family tree of models in that work shows that there is dependence between models, it does not suggest how to account for its effect.

Several studies have defined model independence using a metric that defines scalar distances between different model simulations. Abramowitz and Gupta (2008) proposed constructing a projected model distance space by defining pairwise model distances as the overlap of probability density functions (PDFs) of modelled variables between model pairs. Model behaviour was clustered using self-organising maps (Kohonen, 1989), and the overlap of model output PDF pairs for each cluster was determined. PDF overlap at each cluster was then weighted by the occurrence of cluster conditions to determine

model-model distances. Sanderson et al (2015a) proposed constructing a projected model distance space by defining pairwise model distances using the rows of an orthogonal matrix of model loadings in the singular value decomposition of seasonal



climatological anomaly values of a range of climate variables. Knutti et al (2017) and Lorenz et al (2018) used pairwise root mean square distances between model simulations in one or more variables to assess dependence.

All three of these approaches allow the definition of distance between different model simulations and observational datasets

of commensurate variables to be defined. None verified that the space created met formal metric space criteria in the mathematical sense. That is, they might in theory violate the *triangle inequality* ($d(a,c) \leq d(a,b) + d(b,c)$), where $d(a,b)$ defines the distance between models *a* and *b*, or the *identity of indiscernibles* ($d(a,b)=0$ if and only if $a=b$), for example, and describing these measures as 'distances' could then potentially be misleading. It is unclear whether these potential issues arise or are relevant in practice.

Inter-model distances may also be problematic as measures of independence because they are holistic. That is, inter-model distances reflect the combined effect of *all* process representations that affect the chosen metric - including both those processes strongly supported by observational data and those where a lack of observational data might allow a departure from the true system behaviour (where we might only want to define independence in terms of the latter, as noted above). Further, by

examining model output that is the result of the *interaction* of all of these process representations (that is, just the impact variable values in model output), they ignore the possibility that their combined effect might lead to equifinality (a case of the *identity of discernibles* being violated). That is, different models may arrive at very similar impact variable values through different mechanisms and feedbacks, and so may inappropriately be deemed to be dependent, although this is less likely as the dimensionality of a metric increases. This can to some extent also be tested in cases where models are known to share important

components.

On the other hand, a strength of the three inter-model distance techniques is that an observational estimate can effectively be considered as just another model. Specifically, they allow us to measure distances between different observational estimates and compare them to inter-model distances, and perhaps visualise this in a low dimensional projected space (as shown in

Figure 1).

There is also a growing amount of work in the statistical literature that has not been applied to large model ensembles such as CMIP, but to conceptually related problems, that could more comprehensively define model-model and model-observation similarities (e.g. Chandler et al., 2013; Smith et al., 2009). These approaches view model outputs and observations as

realizations of spatial, temporal or spatio-temporal random processes, and can hence make use of the powerful framework of stochastic process theory. Combined with a Bayesian implementation, these approaches could in principle address many of the shortcomings of approaches such as those above, like the need to use a single metric or to apply a single weight for each model or simulation. They also could allow for more comprehensive and transparent uncertainty quantification as part of the formulation of dependence. There are, however, practical questions related to computation and details of the application, which

are yet to be addressed.

## 6 Independence and performance

Figure 1 also highlights why an assessment of model independence can be at least partly conditional on model performance information. Suppose for example that a 'radius of similarity' were used to identify dependent models in one of the inter-model distance spaces defined in the section above, as illustrated in Figure 1 by the red shaded regions around models (this

idea is raised in both Abramowitz, 2010 and Sanderson et al, 2015a; 2015b). In Figure 1a, Models 1 and 4, by virtue of being relatively close together might be deemed dependent, and so somehow down-weighted relative to models 2 and 3.



In Figure 1b the model positions are identical but observational datasets now lie between models 1 and 4, making the picture less clear. Models 1 and 4 both appear to perform very well (since model-observation distances are relatively small), and since they are spread around observational estimates, might be considered to be independent. In this sense, inter-model distances alone in the absence of observational data are an incomplete proxy for model independence. Both Abramowitz and Gupta

(2008) and Sanderson et al (2015a, 2015b) recognise this issue and try to address it by proposing model independence weights that scale cumulative model-model distances for an individual simulation by its proximity to observations. Neither explicitly addressed how multiple observational estimates might be incorporated, but there is also no theoretical barrier to this.

An alternative approach that combines model distance and performance information is to define model dependence in terms

of model error covariance or error correlation (e.g. Jun et al, 2008a; Jun et al, 2008b; Collins et al, 2010; Bishop and Abramowitz, 2013). This has the advantage that 'error' only reflects *deviations* from an observational product (rather than similarity in model outputs per se), and while it still suffers from the integrative holism noted in the section above (that is, we are only assessing the integrated effect of all process representations), differences in the structure of error between models are likely to reflect differences in the sections of model representation that are *not* tightly constrained by observations.

Incorporating different observational estimates in this case, unfortunately, is more complicated.

A little thought about the values of error correlation that we might expect between independent models reveals how problem-dependent accounting for model dependence can be. If, for example, we examine gridded climatological (time average) values of a variable of interest, then under the (flawed) assumption that an observational estimate is perfect, and the period in question

is stable and long enough to define a climatology, departures from observed climatology might reflect a model's inability to appropriately simulate the system and so represent epistemic uncertainty. In this case, we might suggest that independent simulations should have pairwise zero error correlation, as is the case for independent random variables, since we might *a priori* believe climatology to be deterministically predictable (that is, that a perfect model should be able to match observations). Just as the mean of $n$ uncorrelated random variables with variance 1 should have variance 1/n, we should expect

that the ensemble mean of independent models defined in this way would (a) perform better than any individual simulation, and (b) asymptotically converge to zero error as the size of the ensemble of independent models (with zero error correlation) increases. This is illustrated in Figure 2a, which shows 30 yellow lines, with each comprising of 50 draws from the normal distribution $N(0,0.125)$. Each of these lines is a conceptual representation of error from a different model, where zero error would be shown as the horizontal black line (imagine, for example, that the horizontal axis represented different points in

space). The red line shows the mean of these 30 model error representations, and clearly has significantly reduced error variance. This understanding of independence within an ensemble has been dubbed the "truth-plus-error" paradigm (see Annan and Hargreaves, 2010; 2011; Knutti et al, 2010b; Bishop and Abramowitz, 2013; Haughton et al, 2014; 2015), and has often been assumed rather than explicitly stated (e.g. Jun et al, 2008a; 2008b).

**7 Independence and aleatory uncertainty**

But the truth-plus-error framework is not always appropriate. Knutti et al (2010b) noted that the ensemble mean of the CMIP3 ensemble did not appear to asymptotically converge to observations as ensemble size increased. Gleckler et al. (2008) also show that the variability of the ensemble mean is much less than individual models or observations – and so does not represent a potentially real climate. If we wish to consider ensemble simulations where unpredictability or *aleatory* uncertainty is an inherent part of the prediction, we no can longer maintain an expectation that the system might be entirely deterministically

predictable. This includes, for example, any time series prediction where internal climate variability between models and observations is out of phase (e.g. CMIP global temperature historical simulations or projections from 1850 initialisation), or climatology (mean state) prediction where the time period is too short to be invariant to initial state uncertainty. In these cases



we accept that some component of the observational data is inherently unpredictable, even for a perfect model without any epistemic uncertainty. Ensemble spread in this case might ideally give an indication of the amount of variability we might expect from the chaotic nature of the climate system given uncertain initial conditions, and could be investigated using initial-condition ensembles of climate change projections (Kay et al. 2015, Deser et al. 2016) as well as in the context of numerical

5       weather prediction ensembles (e.g. Hamill et al, 2000; Gneiting and Raftery, 2005).

A simple illustration of the role of aleatory uncertainty is shown in Figure 3, taken from the Technical Summary of WG1 in the Intergovernmental Panel on Climate Change (IPCC) Fourth Assessment Report. Internal variability within each climate model simulation (yellow lines) and observations (black line) is out of phase, so that the variance of the multi-model mean

(red line) is significantly less than individual models or the observations. While ensemble spread in this case represents a combination of both epistemic and aleatory uncertainty, it should be clear that the lack of predictability caused by internal variability removes the expectation that the model ensemble should be centred on the observations.

A synthetic example illustrates this point. If we assume that observations of global mean temperature anomalies in Figure 3

are well approximated by the sum of a linear trend and random samples from $N(0,0.125)$ - the black line in Figure 2b - then an ensemble of independent models that adhered to the truth-plus-error paradigm might look like the yellow lines in Figure 2b. Each of these are the same 'models' shown in Figure 2a, but this time they are presented as a time series and shown as random deviations about the 'observations' (rather than the zero line; 'models' are shown rather than model error). It is perhaps no surprise in this situation that the mean of this 30 member ensemble (the red line) very closely approximates the 'observations'.

This is clearly very different to the role of the ensemble mean we see in Figure 3 (red curve).

An alternative to the truth-plus-error paradigm is to consider observations and models as being draws from the same distribution (e.g. Annan and Hargreaves, 2010; 2011; Bishop and Abramowitz, 2013; Abramowitz and Bishop, 2015). Figure 2c shows the same 'observations' as Figure 2b, but this time represents models in the same way as observations - the deviations

from the linear trend (instead of deviations from the observations, as in 2b). In this case we can see that the ensemble mean (again in red) has much lower variability than observations, as seems evident for the first half of the 20th Century in Figure 3. By introducing external forcing common to the representations of models and observations in Figure 2 - three step deviations that gradually return to the linear trend, intended to approximate volcanic forcing at the locations shown by grey lines in Figure 2d - we can produce an entirely synthetic ensemble that very closely approximates what is shown in Figure 3.


There are of course many reasons why what is shown in Figure 2d is not an appropriate representation of models or observations. Collections of simulations such as CMIP are in reality a mix of epistemic and aleatory uncertainty, not just the aleatory uncertainty shown in Figure 2. The nature of the perturbation that results from external forcing (such as the faux volcanoes in Figure 2d), as well as the nature of internal variability itself, are also likely functionally dependent upon forcing

history, and models exhibit different trends. Nevertheless this simplistic statistical representation of ensemble spread closely approximates the nature of the CMIP ensemble.

Bishop and Abramowitz (2013) argued that independent climate simulations should have the statistical properties of the 'models' in Figure 2d. Specifically, since error-free observations and perfect models (i.e. without epistemic uncertainty) would

both be draws from the same distribution (they named samples from this Climate PDF 'Replicate Earths'), they should both approximate the same level of variability about the mean of this distribution (which represents the forced signal), given enough time. They attempted to partially account for epistemic uncertainty in the CMIP ensemble by offering a transformation of models so that the transformed ensemble strictly adhered to two key statistical properties of this distribution defined entirely



by aleatory uncertainty. These were, given a long enough time period, that (a) the best estimate to any particular replicate Earth is the equally weighted mean of a collection of other replicate Earths (that is, CPDF mean), and (b) the time average of the instantaneous variance of this distribution (the CPDF) across replicate Earths should approximate the variance of any individual replicate Earth about the CPDF mean over time. By treating observations as the only true replicate Earth, they
transformed the CMIP ensemble to be replicate Earth like, with respect to these two properties.

Annan and Hargreaves (2010) also proposed that observations and models are best considered as draws from the same distribution. The meaning of ensemble spread in the 'statistically indistinguishable' paradigm they propose, however, is not immediately clear, and is not explicitly stated in Annan and Hargreaves (2010). They do not discuss internal variability, but in
a later blog post suggested spread represented "collective uncertainties about how best to represent the climate system", which seems to imply epistemic uncertainty. Both Annan and Hargreaves (2010) and Bishop and Abramowitz (2013) suggested that pair-wise error correlation between independent model simulations should be 0.5, since the observations are common to both.

Thus categorical separation of epistemic and aleatory uncertainty is challenging since it requires an accurate quantification of
internal variability. While we have some tools that can help us estimate internal variability, ultimately we have measurements of just one realisation of a chaotic Earth system, and internal variability is affected by the state of the Earth system and forcing conditions (Brown et al. 2017). There is also evidence that the internal variability in some modelling systems (i.e. initial conditions ensembles - see Collins et al. 2001; Deser et al. 2012) may not be a good representation of internal variability in the climate system (e.g. Haughton et al, 2014; 2015). Each of the techniques that give an indication that ensemble spread is
similar to internal variability, such as rank histograms (Hamill, 2001), spread-skill scores in forecasts, Brier Skill Score (Brier, 1950; Murphy 1973), reliability diagram (Wilks, 1995), also have the potential for misinterpretation (e.g. regional biases in an ensemble appearing as under-dispersiveness). In addition, timescales of internal variability are difficult to ascertain from our sparse and short observational record, but there is some evidence that it may operate on very long time scales (e.g. James and James, 1989; Ault et al, 2013; PAGES 2k Consortium, 2013). So while we have techniques for assessing and accounting for
model dependence of epistemic uncertainty that try to nullify aleatory uncertainty by averaging over time, the potential for unquantified aleatory uncertainty to compromise this strategy remains real.

**8 Robust strategies for calibration**

Given that *a priori* measures of independence have yet to prove robust and that aleatory uncertainty could confound the ability
to interpret model-observation distance as purely epistemic uncertainty, how might proposals to account for independence interpreted? Recent experience suggests caution: accounting for dependence or performance differences within an ensemble can be very sensitive to the choice of variable, constraining observational data set, metric, time period and the region chosen. Herger et al (2018a), for example, detail an approach that optimally selected subsets of an existing ensemble for properties of interest, such as the root mean square (RMS) distance of the sub-ensemble mean from observations of a variable's climatology.
The resulting subsets are sensitive to nearly every aspect of the problem including which variables are considered; whether the weighting is inferred from climatological fields, time and space variability, or trends; and whether subsets are chosen before or after bias correcting model projections. Even for a given variable notably different sub-ensembles are obtained when using different constraining observational estimates, even for relatively well-characterized quantities such as surface air temperature.

These types of results are familiar to researchers who utilise automated calibration techniques, and reinforce that post-processing to account for dependence or performance differences within an ensemble, whether by weighting or sub-selecting ensemble members, is essentially a calibration exercise. It also reinforces that thorough out-of-sample testing is needed before



one might be confident that weighting or ensemble sub-selection will improve climate projections or an impacts assessment. It is clear that most post-processing approaches improve ensembles as intended using the data used to derive them (that is, they work well in-sample, typically using historical data). But how can we have confidence that this is relevant for the projection period?

In the context of climate projections we have at least two mechanisms to assess whether the data and experimental setup used to derive the weights or ensemble subset provide adequate constraint for the intended application. The first is a traditional calibration-validation framework, where available historical data is partitioned into two (or more) sets, with the first used to calibrate weights or a sub-ensemble, and the second used to test their applicability out-of-sample (e.g. Bishop and Abramowitz,

2013). For most regional to global climate applications, this will often be limited to 60 (or likely fewer) years of quality observational data – depending on the region and variable – and if the intended application period is far enough into the future, the nature of climate forcing in the calibration and validation periods might not be sufficiently representative for the application. For some quantities, paleoclimate records also offer the potential for calibration-validation testing but their application to weighting and/or subselection has been limited (see Schmidt et al, 2014).

The second approach is model-as-truth, or perfect model experiments (e.g. Abramowitz and Bishop, 2015; Sanderson et al, 2017; Knutti et al, 2017; Herger et al, 2018a, Herger et al, 2018b). This involves removing one of the ensemble members and treating it as though it were observations. The remaining ensemble is then calibrated (that is, weighted or sub-selected) towards this 'truth' member, using data from the historical period only. The calibrated ensemble can then be tested out-of-sample in

the 21st century since the 'truth' member's projections are known. The process is repeated with each ensemble member playing the 'truth' role, and in each case, the ability of the sub-selection or weighting to offer improvement over the original default ensemble is assessed. In the case of weighting, the weighted ensemble could be compared to the equally weighted original ensemble mean; in the case of ensemble sub-selection, comparison could be with the entire original ensemble, or a random ensemble of the same size as the subset. Results are synthesised across all model-as-truth cases to gain an understanding of

the efficacy of the particular approach being tested.

Note that for this approach to have relevance, the similarity of ensemble members needs to be approximately equal to the similarity between observations and ensemble members, in metrics that are relevant to the calibration process and application. For example, if a model-as-truth experiment were performed using all CMIP ensemble members, including multiple initial

conditions members from the same model, the ensemble calibration process could fit the 'truth' simulation much more closely than models are likely to be able to fit observational data. That is, weighting or sub-selection would favour any simulations from the same model as the truth ensemble member, so that the experiment's success might be misleading. This suggests eliminating obvious duplicates before the perfect model tests (see e.g., Figure 5 in Sanderson et al, 2017). It is also worth emphasising that the motivation for this process is not to test the weights or ensemble subset as far out-of-sample as possible,

but rather to ensure that the calibration process is appropriate for its intended application. Note that biases shared among models, especially those which affect projections, will increase agreement among models relative to observations, so that model-as-truth experiments should be treated as a necessary but not sufficient condition for out-of-sample skill.

Thorough out-of-sample testing is important for a number of different reasons. The first, and perhaps most obvious, is to insure

against overfitting due to sample size. We need to ensure that the weighting or sub-selection approach we use has been given enough data to appropriately characterise the relationship between the models we are using, especially if there are many of them, and the constraining observational data. A naive rule-of-thumb for any simple regression problem is roughly 10 times the number of data points as there are predictors (models in this case). While covariance between data points can complicate





this rule, it should give an indication of whether any poor performance in out-of-sample testing is simply due to a paucity of observational data.

A second reason is *temporal transitivity*, making sure that the time period and time scale used to calibrate the weights or ensemble subset provides adequate constraint on the intended application period and time scale. For example, Herger et al (2018a) found that selecting an ensemble subset to minimise climatological surface air temperature bias in the historical period (1956-2013) provided good out-of-sample performance in 21st century (2013-2100) model-as-truth experiments. When this was repeated using linear surface air temperature trend instead, good in-sample improvements were not replicated out-of-sample. That is, the biases in climatology had high temporal transitivity, or predictability out-of-sample, while the biases in

trend did not. This example illustrates why temporal transitivity is particularly important in the case of future projections. It is possible to have two models that have similar behaviour in current climate, for example because the models have both been developed with the same observational datasets for comparison, yet have very different climate sensitivities. As well as temporal period transitivity, one might also consider transitivity between one time scale and another (e.g. the relevance of calibration using monthly data for daily extremes).

The third aspect of out-of-sample testing to consider is *metric transitivity*. That is, ensuring that the metric used to weight or sub-select the ensemble constrains the quantity of interest that the ensemble will ultimately be used for. There are many examples of published work where metric transitivity was simply assumed. Abramowitz and Bishop (2015) assumed that historical RMS distance in gridded time and space fields of surface air temperature and precipitation informed global mean

temperatures and end-of-21st-century projections. Sanderson et al (2015) assumed that optimising for dependence in a multivariate seasonal climatology provided a constraint on climate sensitivity. There is of course no *a priori* reason why these assumptions should be valid, and indeed they could be tested with appropriate model-as-truth analysis.

Given observational constraints, one could make similar arguments that *spatial transitivity* might be another aspect of out-of-

sample testing that is of relevance, depending on the particular application. This could apply both to calibrating using one region and testing at another, and calibrating at one spatial scale and applying at another. For example, Hobeichi et al (2018) tested the ability of the weighting approach outlined in Bishop and Abramowitz (2013) to offer performance improvements in locations not used to derive weights, as well as the ability of site-scale measurements to improve global 0.5° gridded evapotranspiration estimates. One might also evaluate CMIP simulations at coarse spatial scales as a way of deciding which

subset of simulations to regionally downscale, making the assumption that low resolution performance will translate to downscaled performance.

The need for out-of-sample testing applies equally to the use of bias correction techniques (e.g. Macadam et al, 2010; Ehret et al, 2012), and indeed to the entire chain of models used in downstream climate applications (e.g. Clark et al. 2016), although

there are very few examples of this being done.

## 9 Towards generalised ensemble calibration

The sensitivity of weighting and sub-ensemble selection results to metric, variable, observational estimate, location, time and spatial scale and calibration time period underscores that a characterisation of model dependence is not a *general* property of

an ensemble, but application specific. Dependence is not a property of a model simulation per se, rather a property of a specific quantity in a particular simulation with respect to the rest of an ensemble. Nevertheless, in instances where a very specific



variable and cost function are known to be the only properties of interest, it is quite likely that, with appropriate out-of-sample testing regime, a solution to improve projection reliability can be found using existing techniques.

If, however, we decided that application-specific calibration were not generally satisfactory, and that we wanted to try to
calibrate a given CMIP ensemble for model dependence *without* knowing the intended application, how would we do this in a way that would be defensible? Given the increasing number and range in quality of CMIP contributions, it might be useful to suggest a strategy for *general* ensemble pre-processing for a range of applications.

We propose that using one model from each modelling institution that submitted to CMIP is the best general-purpose selection
strategy. This strategy has proved a reasonable approximation to more detailed quantitative approaches that account for model dependence in the CMIP5 ensemble (Abramowitz and Bishop, 2015; Leduc et al, 2016b). This 'institutional democracy' approach requires two important caveats, namely care in excluding models that are near-copies of one another submitted by different institutions and equal care in including models from the same institution with significantly different approaches or assumptions. Given due diligence institutional democracy is a simple but reasonably effective approach to accounting for
model dependence which, we argue, provides a better basis on which to calculate a naive multi-model average for generic purposes such as projection best estimates in IPCC reports.

Institutional democracy is an *a priori* approach is not bound by any particular statistical metric, variable or observational estimate. As institutions increasingly copy whole models or components, however, there is no guarantee that such an approach
will remain effective in the future.

The approach is similar in spirit to one proposed by Boé (2018) to account for the number of shared components by GCMs. As noted above, Boé's approach quickly becomes difficult and time consuming for large ensembles such as CMIP, given the lack of transparency regarding precisely what constitutes different models and the role of tuning. Using this information to
account for dependence would also likely be difficult, since categorical inclusion or exclusion of simulations seems the only option. Also we note that shared history as it pertains to dependence should only include process representations that are not tightly observationally constrained (so that Navier-Stokes equations might *not* represent dependent process treatment, for example), as discussed above - model convergence might well imply accuracy, rather than dependence.

A more comprehensive *a posteriori* approach to generalised calibration might be to simultaneously optimise for all the variables, metrics and observational estimates believed to be informative. The simplest way to do this is to combine all the relevant cost functions into a single cost function for optimisation, resulting in a single optimal ensemble, or set of weights for model runs in the ensemble. This solution, while tractable, has at least two potential disadvantages. First, it makes assumptions about the relative importance of each term in the final cost function that are hard to justify. Different variables and metrics
have different units, so these need to be standardised in some way. Given the shapes of the distributions of different variables can be very different, the approach to standardisation will impact the nature of the final weights or ensemble sub-selection.

Second, this approach risks underestimating uncertainty. If calibration against one key cost function (for example surface air temperature climatology) gives a very different ensemble subset or weights to calibration against another (say precipitation
extremes), the discrepancy between these optimised outcomes is important information about how certain our optimal estimates are. If, for example, there were a universally independent subset within the larger ensemble, it would be the same subset for both variable optimisations. The discrepancy between them is an indication of the degree to which our model ensemble is not commensurable with the observations of the climate system we are trying to simulate. As noted above, one



way to try to minimise overconfidence (underestimated uncertainties) is to use model-as-truth tests to test for variable and metric transitivity, although this cannot avoid the issue of shared model assumptions.

These difficulties may be avoided if we take a more expansive view of what optimization means. A broader approach could use multiple criteria separately (e.g. Gupta et al, 1999; Langenbrunner and Neelin, 2017) in the context of ensemble sub-selection. One might, for example, examine the spread of results from both the surface air temperature climatology optimal ensemble *and* the precipitation extremes optimal ensemble when considering use of either variable. More generally, we propose that the optimal ensembles obtained from optimising against each relevant dataset-variable-metric combination of interest are all effectively of equal value. A generalised ensemble calibration would utilise *all* of these ensembles for projection - that is, use an ensemble of ensembles - and ideally use a Pareto set of ensembles. This would give a far better description of the uncertainty involved with projection, since uncertainty due to models' inability to simultaneously simulate a range of aspects of the climate system can now be expressed as uncertainty in a single variable and metric. This remains a proposal for thorough exploration at a later date.

## 10 Recommendations and next steps

As we have discussed it is unlikely that model dependence can be defined in a universal and unambiguous way. In the absence of easy and agreed-upon alternatives, many studies still use the traditional "model democracy" approach; and indeed it seems unlikely that a "one size fits all" approach or list of good and independent models would be meaningful. However, we argue that users do have a suite of out-of-sample testing tools available that allow the efficacy of any weighting or sub-sampling approach to be tested for a particular application. These should be applied to understand the validity both of the technique itself, and the intended application in terms of metric, temporal and spatial transitivity, as discussed above. If out-of-sample results are robust, they give an indication of the degree to which dependence affects the problem at hand, by way of contrast with the equally weighted status quo. As many of the studies discussed above have shown, accounting for dependence can give markedly different projections. This is particularly important for the upcoming CMIP6, where the model dependence issue is expected to increase. Even the somewhat naive approach of institutional democracy is likely to be a less biased approach to ensemble sampling. Nonetheless we discourage the use of weighting or sub-sampling without out-of-sample testing, as the risks may well outweigh the potential benefits (Weigel et al, 2010; Herger et al, 2018b).

For most applications, questions of how best to select physically relevant variables, domains and appropriate metrics remain open. This should ideally be done by considering the relevant physical processes for the phenomena in question. For the cases where a specific variable and scale is clear, a comparison of existing approaches, discussion of the circumstances in which they should be used and construction of an appropriate out-of-sample testing regime would help guide users' choices. If a more holistic ensemble calibration is needed, further exploration of the idea of multi-objective optimisation is required, which results in the novel concept of an ensemble of model subsets or weighted averages.

There has also recently been a push for more transparency regarding models' development history. This is relevant for the *a priori* approaches, which are based on similarity of model codes. In terms of those approaches, there is also a need to explore the impact of tuning on dependence (Schmidt et al, 2017; Hourdin et al, 2016). It has been shown that parameter perturbations based on otherwise identical code bases (such as in the climateprediction.net exercise; Mauritsen et al, 2012) can lead to notably different projections. Better documentation from the modelling institutions regarding the standard metrics used to judge a model's performance during development and the preferred observational products used for tuning is needed. This information can help determine the effective number of independent models in an ensemble in relation to the actual number of models for a given application.



We stress again that the simulations made for CMIP do not represent a designed ensemble. In particular the simulations do not span the full uncertainty range for GCM projections or systematically sample the set of all possible model configurations. This is something to keep in mind for any subset-selection or weighting approach. Moreover, it is currently unclear how to deal

with the situation when models start to converge on the true climate state, which might occur as models resolve more and more processes. In such a situation, despite considering them to be interdependent, we do not want to eliminate them. This might get even more complex when dealing with observational uncertainty.

Model-as-truth analyses are essential to test the skill of any weighting or sub-setting approach out-of-sample. However, they

are necessary-but-not-sufficient tests and have the potential for overconfidence given that many climate models are based on similar assumptions and are thus not truly independent. While some steps can be taken to ameliorate this issue, due to the central role of such analyses, the limits of these tests should be explored in future studies and guidelines provided regarding how to best set them up.

## 11 Conclusions

With model component and process representation replication across nominally different models in CMIP5, and the anticipation of more to come in CMIP6, the need for an effective strategy to account for the dependence of modelled climate projection estimates is clear. Perhaps the biggest obstacle to doing this is that the manifestation of model dependence is problem-specific, so that any attempt to address it requires an approach tailored to individual projection impact analyses. We presented a holistic framework for understanding the diverse and apparently disparate collection of existing approaches to

addressing model dependence, noting that each addresses slightly different aspects of the problem.

Critically, we reinforce that the efficacy of any attempt to weight or sub-select ensemble members for model dependence or performance differences, or indeed bias correction, must be tested out-of-sample in a way that emulates the intended application. Calibration-validation with different time periods within the observational record, as well as model-as-truth

experiments were discussed as two approaches to doing this.

Universal calibration of an ensemble for model dependence that is not specific to a particular application remains elusive. In that context, pre-selecting simulations based on an *a priori* knowledge of models, and in particular using institutional democracy (one model per institute, with additional removal of any supplementary simulations that are sourced from known

model replicates at different modelling institutes and/or the addition of clearly distinct variants from within a single institution) is more defensible than naive use of all available models in many applications.

The final step of relating dependence in model output to similarities in model structure can only be achieved once we have a transparent system for documenting and understanding the differences in treatment of processes, and tuning, between different

climate models. While there are some ad-hoc examples of attempts to do this (e.g. Figure 5 in Edwards, 2011; Masson and Knutti, 2011), a formal requirement to document the nature of model structure, parameter evolution, and freely available source code would be a welcome step that would spawn new areas of enquiry in this field (Ince et al. 2012). This would ultimately result in more effective investment in model components that provide independent projection information and bring the community a step closer to producing well calibrated ensembles for climate projection.



**Acknowledgements**

We wish to thank the Australian Research Council Centre of Excellence for Climate System Science for funding the workshop in December 2016 at the National Center for Atmospheric Research (NCAR) that seeded this paper, as well as the NCAR IMAGE group for hosting the workshop and providing administrative support. NCAR is sponsored by the US National Science
Foundation. Thanks also to The World Climate Research Program Modelling Advisory Council that endorsed the workshop. Extensive discussions at a second workshop in July 2017 at the Aspen Global Change Institute on *Earth System Model Evaluation to Improve Process Understanding* also contributed significantly to the content of this work. Gab Abramowitz is supported by the ARC Centre of Excellence for Climate Extremes (CE170100023) and Nadja Herger by the ARC Centre of Excellence for Climate System Science (CE110001028). Ruth Lorenz is supported by the European Union's Horizon 2020
research and innovation program under grant agreement 641816 (CRESCENDO). Tom Hamill, James Annan and Julia Hargreaves provided valuable feedback.

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





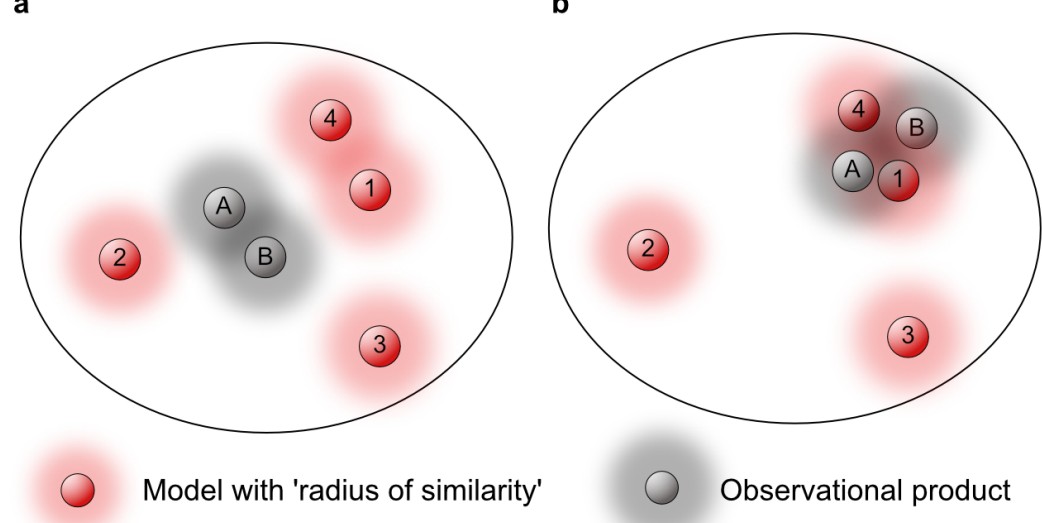

**Figure 1: A two-dimensional projection of an inter-model distance space, showing different models and observational estimates, with a radius around models that could be used to determine model dependence. The radius around observations might be related to the uncertainty associated with a given observational estimate. The two panels illustrate how the relative position of observational data sets in this space could complicate this definition of model dependence.**



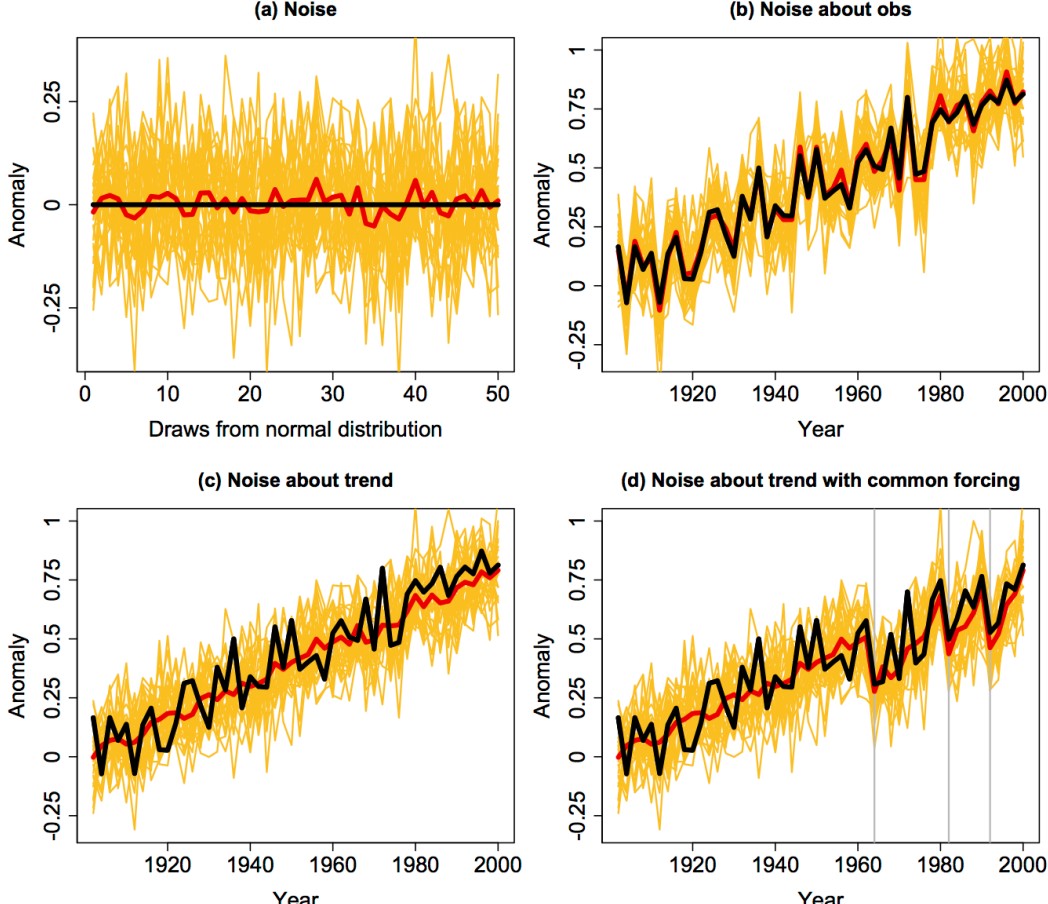

**Figure 2:** (a) The 'truth-plus-error' paradigm illustrated using random samples from *N(0,0.125)* (yellow lines) as a proxy for error in models in an ensemble, with 'observations' in black and the multi-model mean in red. (b) The same 'models' shown as deviations from 'observations' approximated by a noisy linear trend. In contrast, panel (c) illustrates model error and observation time series as draws from the same distribution, both shown as noise about a linear trend, and (d) the effect of applying common forcing perturbations to both models and observations. The same draws from *N(0,0.125)* are used in all four panels.



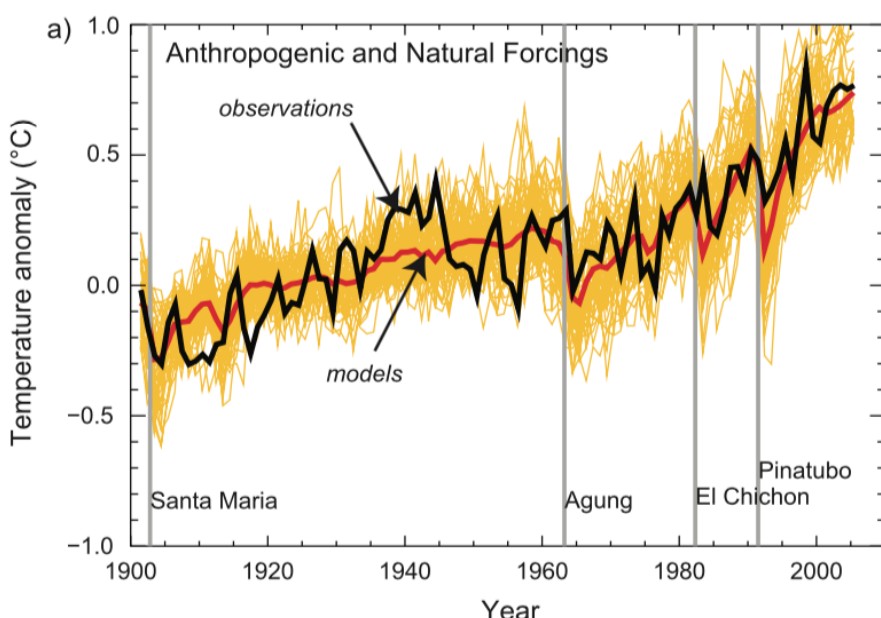

**Figure 3:** TS.23 taken from the Technical Summary of WG1 in the IPCC Fourth Assessment Report, showing the multi-model mean's decreased variability (red) relative to individual models (yellow) and observations (black), as well as the effect of volcanic forcing on ensemble behaviour.