# Peer review of "Model dependence in multi-model climate ensembles: weighting, sub-selection and out-of-sample testing"

_Earth System Dynamics, 2018_

## Referee Comment (RC1) · Anonymous Referee #1 · 7 Aug 2018

Review Comments: The authors present a review of a wide array of studies that address model dependence in multi-model climate ensembles. They also provide valuable guidance on how users can test the efficacy of approaches that account for dependence. This is a very timely contribution coming as it does in the months leading up to CMIP6 –easily the largest climate modeling exercise ever carried out. This manuscript will set the stage for how model simulations are treated – and hopefully move beyond equally weighted simulations.

My main comment is that the manuscript needs to be better organized. It currently has 11 sections and no roadmap up front on what these sections contain. Surely, there is a

way to consolidate some of these into subsections and have fewer main sections. Most of the comments below are also related to readability.

Detailed comments: 1. Page 2, Line 8: The six sources of uncertainty are in the first paragraph.

2. Page 2, Line 25: The word "calibrated" may be replaced with "constructed" or "designed".

3. Page 2, Line 36: Again the word calibrated is used in a different sense than is standard.

4. Page 4, Line 6 and Line 29: The analogies from evolutionary classifications may not help the typical reader or even make a connection. I suspect many, like this reviewer will have to look it up.

5. Page 4, Line 20: The use of "first version number" and "second version number" are actually better understood as MajorRevisionNumber and MinorRevisionNumber. Boé (2018) provides a version number example that avoids the confusion in this manuscript. Perhaps move the CLM4/CLM4.5 example up to avoid this confusion?

6. Page 4, Line 24-26: "A reminder . . .." is a complicated sentence. Perhaps rewrite in simpler language.

7. Page 5, Line 32: In the sentence ". . .these approaches could in principle address many of the shortcomings of approaches such as those above. . . " I believe the latter "approaches" refers to the 3 discussed in the preceding paragraphs. This sentence should be reworded to make it clear.

8. Page 6, Lines 1-7: Is there an implicit assumption (in figure 1) that observational estimates are close together? In the regional context, it is quite common for multiple observational estimates to be further apart than inter-model distances.

9. Page 8, Line 28: The word "calibration" appears for the first time in the section

heading. Some prior context is important.

10. Page 8, Line 30: "... how might proposals to account for independence interpreted?" is missing a "be".

11. Page 9, Line 6: "data" here refers to "observed data" right? This needs to be clarified.

12. Page 10, Lines 40-41: The sentence "Dependence is not a property of a model simulation per se, rather a property of a specific quantity in a particular simulation with respect to the rest of an ensemble" is a variant of the immediately preceding sentence.

---

## Referee Comment (RC2) · Anonymous Referee #2 · 7 Oct 2018

The authors provide a very timely and valuable review of recent studies that have tried to cope with the issues related to model dependence and performance in multi-model climate ensembles. They make clear that the climate community needs to go beyond model democracy "one model, one vote" in the future use of CMIP projects for climate and impact studies (and I agree very much with that statement). They also provide some guidance on the testing and evaluation of methods that are commonly used to move beyond equally weighted ensembles.

I fully support the publication of the paper pending some minor revisions described below.

Section 1: this section is an important one as it seeks to define the different types of uncertainty involved in climate projections. I find the current text a bit confusing. For instance, the paper begins with a list of uncertainty sources (six of them are mentioned) and then proposes to classify them in the second paragraph in two main classes: epistemic and aleatory. They define epistemic uncertainty as our uncomplete "knowledge and understanding of the climate system" implicitly meaning that it is reducible with more information and knowledge. Yet the first source of uncertainty listed in the first paragraph is the uncertainty due to the lack of predictability of human behavior and 21st century history (an uncertainty that includes future GHG emissions but that is indeed much larger). This is usually considered as a third class of uncertainty and is dealt with the use of multiple and plausible scenarios with no explicit credibility ranking among them. Including it in epistemic uncertainty would imply that epistemic uncertainty cannot be reduced to the climate system (as written) but that it has to include the interaction between the climate system and human (social, political and economic) behavior. Another remark is that the limitations in observations are only mentioned as "required for accurate model initialization". Clearly, observations also play a central role in model development and evaluation. Furthermore, the text does not explicitly mention the uncertainty related to the history of past external forcings. Yet this is a major source of uncertainty, in particular at regional scale. As the authors do not begin by their classification, they have to use other terms that are not precisely defined (like structural, page 2, line 3). My suggestion would be to start with the classification right from the beginning (by introducing the three classical types of uncertainty) and then use the same words throughout the text to define the different uncertainty sources. The last paragraph of section 1 states the objectives of the paper but lacks to mention the fact that the sampling GCM strategy is only one piece, albeit an important one, in the complex workflow that goes from emission scenario to climate projection to impact studies. For instance, the independence issue also applies to the design of the GCM-RCMs matrix. Finally, I find a bit strange the last sentence of the section (line 33) as we live in a world where resources for model development are becoming scarce to say
the least while there still are many unresolved modelling issues.

Section 2: Page 3, line 9: there is a need to clearly differentiate throughout the text multi-model ensembles from initial condition ensembles (single-model ensemble with many members differing by their perturbed initial conditions). I suggest avoiding the use of the word member when referring to the former.

Section 3: First paragraph: the authors make an interesting distinction between climate model components/processes where we do not expect epistemic departures from the true physical system (where we expect to have strong dependency among models) and those where we expect to see such departures (and where it would be needed to have independent representations of processes). Yet, I wonder if this distinction is really useful in practice. These components are often tightly coupled (think of the atmospheric dynamics/physics pair of components) in a GCM meaning that errors in the latter would lead to biases in the former (for instance biases in cloud microphysics could lead to biases in temperature gradients that would in turn affect atmospheric circulation). Disentangling the exact origin of the biases in a fully coupled system is a rather difficult task. It would be interesting to propose and discuss the hierarchy of models and experiments that would allow a clean separation between these different types of components/processes. A simple example is the use of SST-forced experiment in addition to a fully coupled one to assess the origin of atmospheric biases.

Section 4: Page 4, lines 16-26: as it currently stands, the text seems to imply that component democracy (instead of model democracy) is too difficult to implement "beyond categorical inclusion or exclusion". I would argue that this exact sentence also applies to the institutional democracy that the authors are advocating for. There is certainly some subjectivity in using version numbers to make claims about independence, but I think there is as much subjectivity in using the modelling center names. If a modelling center has 4 different model versions that differ by physics and/or resolution, the final choice of just one version will also be subjective. Finally, there will also be cases where two modelling centers share most of their components leading to potential strong deInteractive comment

pendency between their simulations. In fact, the authors in their conclusions (page 13, lines 26-31) recognize that some additional work is deeply needed to efficiently use institutional democracy, this extra-work being more or less related to the Boé (2018) type of analysis of the available model documentation and meta-data.

Section 8: Page 9, line 17: one could also cite Boé and Terray (2015) "Can metricbased approaches really improve multi-model climate projections? the case of summer temperature change in France. Climate Dynamics, vol. 45, iss. 7, pp. 1913-1928" which discuss the sensitivity of weighting strategy results to a large range of methodological choices.

Page 10, lines 4-31: what is discussed here has close and strong links with the emergent constraint literature and its recent developments (see for instance Nijsse, F. J. M. M. and Dijkstra, H. A.: A mathematical approach to understanding emergent constraints, Earth Syst. Dynam., 9, 999-1012, https://doi.org/10.5194/esd-9-999-2018, 2018). Yet, there is no mention of it and almost none of the relevant papers is cited. The authors could also discuss (or at least mention) the possible caveats in using regression analysis for the weighting problem: adequacy of the assumed linear model between the predictor and predictand, standard use of OLS instead of TLS (with errorin-variable), sensitivity of the results and selection bias in data pre-processing (like spatial averaging) ...

Page 10, lines 33-35: the authors could also cite some recent references, for instance:

D. Maraun: Bias Correcting Climate Change Simulations - a Critical Review, Curr. Clim. Change Rep. 2:2011-220, 2016.

J.M. Gutiérrez, D. Maraun, M. Widmann, R. Huth, E. Hertig, R. Benestad, O. Roessler, J. Wibig, R. Wilcke, S. Kotlarski, D. San Martín, S. Herrera, J. Bedia, A. Casanueva, R. Manzanas, M. Iturbide, M. Vrac, M. Dubrovsky, J. Ribalaygua, J. Pórtoles, O. Räty, J. Räisänen, B. Hingray, D. Raynaud, M.J. Casado, P. Ramos, T. Zerenner, M. Turco, T. Bosshard, P. Štěpánek, J. Bartholy, R. Pongracz, D.E. Keller, A.M. Fischer, R.M. Car-
doso, P.M.M. Soares, B. Czernecki, C. Pagé. An intercomparison of a large ensemble of statistical downscaling methods over Europe: results from the VALUE perfect predictor crossĂvalidation experiment. Int. J. Climatol., online first, 2018.

E. Hertig, D. Maraun, J. Bartholy, R. Pongracz, M. Vrac, I. Mares, J.M. Gutierrez, J. Wibig, A. Casanueva and P.M.M. Soares: Comparison of statistical downscaling methods with respect to extreme events over Europe: Validation results from the perfect predictor experiment of the COST Action VALUE. Int. J. Climatol., online first, 2018.

D. Maraun, R. Huth, J.M. Gutierrez, D. San Martin, M. Dubrovsky, A. Fischer, E. Hertig, P.M. Soares, J. Bartholy, R. Pongracz, M. Widmann, M.J. Casado, P. Ramos and J. Bedia: The VALUE perfect predictor experiment: evaluation of temporal variability, Int. J. Climatol., online first, DOI: 10.1002/joc.5222, 2017.

Section 9: Page 11, lines 9-16: see comment on section 4 that also applies here.

Page 11, line 19: Institutions also often co-develop (instead of "copy") models and/or components (such as the NEMO ocean engine in Europe).

Page 11, lines 22-28: some of statements in this paragraph are just claims with no supporting evidence ("... quickly become difficult and time consuming ...", "Using this information ... seems the only option"). I think that the issues with regard to component and institutional democracy are quite similar.

Page 12, lines 4-13: the authors might also want to discuss and cite: Borodina, A., E.M. Fischer, and R. Knutti, 2017: Emergent Constraints in Climate Projections: A Case Study of Changes in High-Latitude Temperature Variability. J. Climate, 30, 3655–3670, https://doi.org/10.1175/JCLI-D-16-0662.1

**ESDD**

---

## Author Comment (AC1) · 27 Nov 2018

Please see the supplement PDF for our detailed response to Anonymous Referee #1.

Please also note the supplement to this comment:
https://www.earth-syst-dynam-discuss.net/esd-2018-51/esd-2018-51-AC1-supplement.pdf

---

## Author Response (AR1)

[revised manuscript text omitted]

Alternatively, if we assume that we did have a perfect understanding of how the climate system worked and could codify this effectively in models, climate system stochasticity and chaotic behaviour will also mean that for any given (incomplete) observational record and model resolution, an ensemble of solutions would exist, representing the inherent limit of predictability in the climate system - *aleatory uncertainty*. The distinction between epistemic and aleatory uncertainty is relevant because the nature of model dependence, and so how we might attempt to address it, is different in each case.

The use of multi-model ensembles, including those from the widely-used Coupled Model Intercomparison Project (CMIP), is common in climate science despite the fact that such ensembles are not explicitly constructed to represent an independent set of estimates of either epistemic or aleatory uncertainty. In fact the ensembles are not systematically designed in any way, but instead represent all contributions from institutions with the resources and interest to participate - and so are optimistically called 'ensembles of opportunity' (Tebaldi and Knutti, 2007). The purpose of this paper is to give an overview of approaches that have been proposed to untangle this ad hoc ensemble sampling, to discuss assumptions behind the different methods as well as the advantages and disadvantages of each approach. In the end, the goal is to efficiently extract the information relevant to a given projection or impacts question, beyond naive use of CMIP ensembles in their entirety. While we discuss dependence in the context of global climate model (GCM) sampling here, there are clearly many more links in the chain to impacts prediction, such as regional climate models downscaling, and these issues apply equally to other steps in that chain (see Clark et al, 2016 for more on this). Nevertheless, identifying appropriate approaches for some applications might not only help to give more understanding and certainty to projections and impacts research, but might also help direct limited resources to model development that does not essentially duplicate the information that other models provide
.

In the next section we discuss the nature of sampling in multi-model climate ensembles. In Section 3, we examine which aspects of models we might want to be independent, as opposed to agreeing with observational datasets, and the relevance of canonical statistical definitions of independence. Section 4 details attempts to define model independence in terms of model genealogy, while Section 5 discusses definitions based on inter-model distances inferred from model outputs. We discuss the relationship between model independence and performance in more detail in Section 6, before considering the role of model independence in estimating aleatory uncertainty in Section 7, and how this helps distinguish and contextualise different ensemble interpretation paradigms that are evident in the literature. In Sections 8 and 9, we examine the critical question of how best to test the efficacy of any post-processing approach to address model dependence or performance differences, including weighting or model sub-selection, before making recommendations and conclusions in Sections 10 and 11.

[revised manuscript text omitted]

40   minor revision number was different (for example CLM4 and CLM4.5 would be deemed dependent). However, it is unlikely that the approach to version numbering is consistent across modelling centres, so that two components might be very different even if they share a major version number, or vice versa. Also, it does not seem obvious how we might account for

the effect of shared model histories within an ensemble if we had all this information available, beyond categorical inclusion or exclusion of simulations. As discussed above, an ideal definition of model dependence would only include variability in process representations that are not tightly observationally constrained, so that several models using the Navier-Stokes equations might not represent dependent treatment of process, for example.

**5 Independence as inter-model distance**

[revised manuscript text omitted]